# Comparative Assessment of Docking Programs for Docking and Virtual Screening of Ribosomal Oxazolidinone Antibacterial Agents

**DOI:** 10.3390/antibiotics12030463

**Published:** 2023-02-24

**Authors:** McKenna E. Buckley, Audrey R. N. Ndukwe, Pramod C. Nair, Santu Rana, Kathryn E. Fairfull-Smith, Neha S. Gandhi

**Affiliations:** 1Centre for Genomics and Personalised Health, Queensland University of Technology, Brisbane, QLD 4059, Australia; 2School of Chemistry and Physics, Queensland University of Technology, Brisbane, QLD 4000, Australia; 3Centre for Materials Science, Queensland University of Technology, Brisbane, QLD 4000, Australia; 4Discipline of Clinical Pharmacology, College of Medicine and Public Health, Flinders University, Adelaide, SA 5042, Australia; 5Flinders Health and Medical Research Institute (FHMRI), Flinders University, Adelaide, SA 5042, Australia; 6South Australian Health and Medical Research Institute (SAHMRI), University of Adelaide, Adelaide, SA 5000, Australia; 7Discipline of Medicine, Adelaide Medical School, The University of Adelaide, Adelaide, SA 5000, Australia; 8Applied Artificial Intelligence Institute (A2I2), Deakin University, Geelong, VIC 3220, Australia

**Keywords:** molecular docking, oxazolidinones, antibiotics, rRNA, ribosomes, re-scoring, DOCK 6, Vina

## Abstract

Oxazolidinones are a broad-spectrum class of synthetic antibiotics that bind to the 50S ribosomal subunit of Gram-positive and Gram-negative bacteria. Many crystal structures of the ribosomes with oxazolidinone ligands have been reported in the literature, facilitating structure-based design using methods such as molecular docking. It would be of great interest to know in advance how well docking methods can reproduce the correct ligand binding modes and rank these correctly. We examined the performance of five molecular docking programs (AutoDock 4, AutoDock Vina, DOCK 6, rDock, and RLDock) for their ability to model ribosomal–ligand interactions with oxazolidinones. Eleven ribosomal crystal structures with oxazolidinones as the ligands were docked. The accuracy was evaluated by calculating the docked complexes’ root-mean-square deviation (RMSD) and the program’s internal scoring function. The rankings for each program based on the median RMSD between the native and predicted were DOCK 6 > AD4 > Vina > RDOCK >> RLDOCK. Results demonstrate that the top-performing program, DOCK 6, could accurately replicate the ligand binding in only four of the eleven ribosomes due to the poor electron density of said ribosomal structures. In this study, we have further benchmarked the performance of the DOCK 6 docking algorithm and scoring in improving virtual screening (VS) enrichment using the dataset of 285 oxazolidinone derivatives against oxazolidinone binding sites in the *S. aureus* ribosome. However, there was no clear trend between the structure and activity of the oxazolidinones in VS. Overall, the docking performance indicates that the RNA pocket’s high flexibility does not allow for accurate docking prediction, highlighting the need to validate VS. protocols for ligand-RNA before future use. Later, we developed a re-scoring method incorporating absolute docking scores and molecular descriptors, and the results indicate that the descriptors greatly improve the correlation of docking scores and pMIC values. Morgan fingerprint analysis was also used, suggesting that DOCK 6 underpredicted molecules with tail modifications with acetamide, n-methylacetamide, or n-ethylacetamide and over-predicted molecule derivatives with methylamino bits. Alternatively, a ligand-based approach similar to a field template was taken, indicating that each derivative’s tail groups have strong positive and negative electrostatic potential contributing to microbial activity. These results indicate that one should perform VS. campaigns of ribosomal antibiotics with care and that more comprehensive strategies, including molecular dynamics simulations and relative free energy calculations, might be necessary in conjunction with VS. and docking.

## 1. Introduction

Oxazolidinones are a class of protein synthesis inhibitors that act on the 50S ribosomal subunit across a broad spectrum of Gram-positive bacterial strains, such as *Staphylococcus aureus* (*S. aureus*)*, Enterococcus faecalis (E. faecalis),* and *Streptococcus pyogenes* (*S. pyogenes*), and Gram-negative strains, such as *Escherichia coli* (*E.coli*)*, Pseudomonas aeruginosa* (*P. aeruginosa*)*,* and *Klebsiella pneumoniae* (*K. pneumoniae*). Unlike other 50S inhibitors, such as macrolides, lincosamides, and chloramphenicol, which physically block the initiation or translocation of peptide bond formation between amino acids [1,2], oxazolidinones inhibit protein synthesis at an earlier stage by interfering with the formation of the initiation complex between 50S and 30S subunits [3]. Linezolid, a synthetic antibiotic based on the oxazolidinone scaffold, binds to the 23S peptidyl transferase centre (PTC) of the 50S ribosomal subunit [4]. It stops bacterial growth and reproduction by preventing mRNA and tRNA’s combination with the 50S and 30S ribosomal subunits from forming a 70S initiation complex [3,5,6]. Knowing the mechanism of linezolid binding to the PTC is beneficial for developing new oxazolidinone derivatives. This information can indicate the scope for new drug derivatives and suggest sites for other interactions, which can facilitate the prediction of positive and negative interactions. The structures can suggest drug modifications that will allow it to bind to the ribosome despite resistance factors [7]. Figure 1 demonstrates an example of a detailed three-dimensional ribosomal structure. For the 50S ribosome *S. aureus* (PDB 4WFA, strain NCTC832) crystal structure and linezolid binding site, Figure 1a,b display the protein and ribosomal interactions, and Figure 1c shows the linezolid binding site and structural interactions with residues G2088, A2478, U2533, and U2612 (*E. coli* numbering).

Many studies have investigated structural modifications to the oxazolidinone scaffold in recent decades. A comprehensive overview of oxazolidinone structure-activity relationships (SAR) is detailed by Zhao et al. [6]. This review covers the latest research progress in oxazolidinone antibacterial agents, derivatives with modifications to linezolid and new antibacterial agents containing oxazolidinone backbones [6]. Currently, only two approved oxazolidinones exist on the market, linezolid (1) and tedizolid (2) (Figure 2). Linezolid consists of an oxazolidinone core (A-ring) with a 5-acetamidomethyl side chain (C-5 side chain) and a 3-fluorophenyl ring (B-ring) attached to a morpholino moiety (C-ring). The key structural features of linezolid critical to its activity are the *N*-aryloxazolidinone moiety and the stereochemistry (*S* configuration) at the C-5 position of the oxazolidinone [5]. The morpholino unit has little effect on the activity of the oxazolidinone compound as it does not significantly interact with the ribosomal subunits, but its inclusion results in a better safety profile [9]. The modification of the morpholine ring is prevalent in the literature as it is most tolerant to functionalisation, and a vital example of this is ranbezolid (3). Modification of the C-5 side chain is also common. The replacement of the methyl group in the 5-acetamidomethyl moiety with other small substituents, such as chloromethane (4) and dichloromethane (5), resulted in increased activity (against *S. aureus*, *Staphylococcus capitis* (*S. capitis*), *Staphylococcus epidermidis* (*S. epidermidis*), and vancomycin-resistant *Enterococcus* (VRE, ATCC700221) strains) making these compounds 2-fold more potent than linezolid [6]. While the oxazolidinone core is generally left unchanged, there have been attempts to modify it by developing tricyclic-fused oxazolidinones (6) [6]. Interestingly, the fluorination of the B-ring in these compounds resulted in reduced activity against *S. aureus* (ATCC 29213), methicillin-resistant *S. aureus* (MRSA), methicillin-resistant *S. epidermidis*, penicillin-resistant *Streptococcus pneumoniae* (*S. pneumoniae*), and *Enterococcus faecalis* (*E. faecalis*) strains [10]. In summary, there have been numerous in vitro studies investigating the structural modification of oxazolidinones for drug development and how said modifications affect the activity of the ligand. Another approach to expediate these studies is to use computational chemistry methods, such as molecular docking, to predict how well potential drug candidates bind to a target.

Small-molecule docking or ‘docking’ is a computational approach aimed at predicting the binding poses of a small molecule ligand and biomacromolecules such as protein or nucleic acid (NA) targets such as DNA or RNA. The docking software aims to predict and understand molecular recognition structurally (by finding likely binding modes) and energetically (by predicting scoring) [11]. Traditionally, docking programs, such as AutoDock Vina (Vina) [12], GOLD [13], and Glide [14], were created for protein–ligand targets. Some programs initially developed for protein–ligand docking have now been optimised for NA targets, such as AutoDock4 (AD4) [15], DOCK 6 [16], and FITTED [17], while other docking programs were developed explicitly for NA targets, including NLDock [18], rDock [19], and RLDOCK [20]. Scoring functions are mathematical functions used to approximately predict the scoring between two molecules after they have been docked, in addition to removing bias from a docking algorithm’s internal scoring function. While developed for protein–ligand interactions, there are various programs explicitly developed for nucleic acid–ligand interactions that currently exist. Some docking algorithms with internal scoring functions use force field base (DOCK 6 and RLDOCK) or empirical-based scoring functions (AutoDock and rDock), while standard RNA scoring functions are independent of a docking program and utilise knowledge-based functions (ITScore-NL [21], LigandRNA [22], SPA-LN [23], and DrugScore^RNA^ [24]). In contrast, others utilise machine learning (ML) scoring functions (AnnapuRNA [25] and RNAposers [26]). 

Traditionally, ligand-based approaches such as quantitative structure–activity relationship (QSAR) and pharmacophore modelling have been successfully applied to study the structure-activity relationships of oxazolidinone antibacterials against various strains [27,28,29]. In recent years, docking has been applied to targeting oxazolidinones for hit identification or predicting the interactions and binding pose of a ligand with nucleic acids such as DNA and RNA [30,31,32,33,34,35]. For example, Orac et al. used Glide combined with synthetic and 3D structural approaches to study 4,5-disubstituted oxazolidinones bound to T-box riboswitch RNA [33]. To the best of our knowledge, there has been no rigorous benchmarking study addressing the docking of large oxazolidinone molecule libraries, such as Zhao et al.’s review on 285 oxazolidinone scaffolds [6]. Thus, our goal in this work was to analyse these derivatives against 23S rRNA molecular structures found in ribosomes by conducting an extensive in silico evaluation. 

This evaluation consisted of various computational methods. Molecular re-docking validation was conducted on various docking programs such as AD4 [15], Vina [12], DOCK 6 [16], rDock [19], and RLDOCK [20] to determine how efficient each program was at reduplicating the crystal structures’ native poses and to select the best-performing program to conduct the virtual screening of the dataset. Virtual screening was also performed on the dataset of derivatives, considering the structural modification and MIC activity analysis against *S. aureus*, to determine whether structural modifications or varying MIC activities have an influence on how a derivative performs. Finally, additional methods, such as pharmacophore analysis, re-scoring with external ML scoring functions (AnnapuRNA [25]), and Morgan fingerprint bit analyses, were conducted as supplementary methods, allowing for investigation through ligand-based alternatives. 

## 2. Results and Discussion

### 2.1. Pose Prediction Using Five Commonly Used RNA Docking Programs

We examined five docking programs, i.e., AD4, Vina, DOCK 6, rDock, and RLDOCK, to determine how accurately they reproduce the experimental results by self-docking. The scoring functions and search algorithms of the programs used in this study are outlined in Table 1.

Eleven oxazolidinone ligand-based ribosomal crystal structures with resolutions of 2.0 Å–3.5 Å were selected: 3CPW (2.7 Å), 3CXC (3.0 Å), 3DLL (3.5 Å), 4WFA (3.4 Å), 6DDD (3.1 Å), 6DDG (3.1 Å), 6QUL (3.0 Å), 6WQN (2.9 Å), 6WQQ (3.1 Å), 6WRS (3.2 Å), and 6WRU (3.1 Å). The docking performance when predicting the ligand’s position compared to the original crystallographic conformations varied across the programs (Figure 3a). The performance was ranked based on scoring and root-mean-square deviation (RMSD). Scoring is the strength of the binding interaction between a receptor (protein, RNA, or DNA) and its ligand or small molecule (drug, inhibitor, or derivative). A lower score (i.e., a more negative score) indicates a higher performance, displaying better stability of the receptor–ligand complex. RMSD (measured by Å) is an additional measure of redocking success between programs, which gives the average deviation between the corresponding atoms of two targets. The lower the RMSD, the more similar the native and redocked pose. Docking algorithms have their inbuilt scoring functions and can sometimes be biased towards a ligand depending on the size or the structural components. RMSD is typically used as an analysis technique for comparing a redocked ligand to its native ligand. As the virtual screening (VS) of the derivatives utilized a dataset external from the crystal structure, they did not have a native ligand to compare to, so RMSD was not used for derivatives to compare the whole ligand. The docked poses were considered successful if the ligand RMSD between the docked pose and the structure was less than 2.5 Å [36].

Figure 3b,c demonstrates the average performance of each program across all oxazolidinone-based structures for each program and crystal structure for scoring and RMSD, respectively. Figure 4a,b demonstrates the average minimum, medium, and maximum of the top-performing values for each docking program (the top 5 of each crystal structure averaged) for scoring and RMSD, respectively. The goal of separating the programs into overall and top performing was to show an accurate spread of the performance for each docking program without the separation of specific crystal structures. 

Of the eleven crystal structures tested for AD4, the average binding poses were unsuccessful (Figure 3c), with none of the crystal structures obtaining a value of less than 2.5 Å, with the overall program performance average at 4.8 Å. Additionally, the overall normalized scoring performance was at 0.63, ranking the second lowest of all the docking programs (Figure 3b). The overall average experimental binding pose for Vina was only successful in two cases (Figure 3c). Similar to AD4, it had poor scoring performance in comparison to the other programs. The overall normalized average scoring was 0.63 (Figure 3b). DOCK 6 successfully yielded experimental-like poses in four of eleven complexes, displaying the best sampling power for the overall average values for all the docking program (Figure 3c). It was the highest-performing docking program for overall normalized scoring, being 0.72 on average (Figure 3b). rDock had the lowest top-performing pose with no successful poses for any of the eleven complexes (Figure 3c). rDock performed relatively well regarding scoring, with an overall normalised score of 0.69 (Figure 3b). Overall, RLDOCK did not successfully yield experimental-like poses in any of the eleven crystal structures (Figure 3c). The program had the lowest overall average normalized scoring performance of 0.61 (Figure 3b).

AD4 scored poorly compared to other programs, with a top-performing median normalized value of 0.63 (Figure 4a). AD4 was the second-highest performer for the top-performing median RMSD (Figure 4b); however, only 52% of the top-performing values were successful (being below 2.5 Å), being the third best-performing program (Figure 4c). Vina had a top-performing median scoring of 0.63 and a top-performing median RMSD of 3.5 Å (Figure 4a,b). Only 37% of the top-performing poses scored under the 2.5 Å success score (Figure 4c), with 50% being above 4.5 Å, the highest of all docking programs. DOCK 6 had the highest normalized top-performing median values for both scoring (Figure 4a) and RMSD (Figure 4b), at −0.82 and 1.78 Å, respectively. Moreover, 58% of the top-performing scores were under the 2.5 Å success limit (Figure 4c). rDOCK had a top-performing normalized median value of 0.74(Figure 4a) and a top-performing median RMSD similar to Vina, 3.8 Å (Figure 4b). However, it was the poorest performing in terms of top-performing success rate, with only 10% showing experimental values below 2.5 Å (Figure 4c). RLDOCK had the 3rd lowest top-performing normalized score of 0.66. However, of the top-performing values, RLDOCK was successful in almost 85% of ligand binding poses, being by far the best-performing program in terms of success rate (Figure 4c). For both scoring and RMSD, RLDOCK had the broadest range of top-performing values (Figure 4a,b), most likely due to its “blind docking” of the ligand. Blind docking refers to the docking of a ligand to the whole surface of a protein or RNA target without defining the ligand’s active site. As RMSD measures the average deviation of the molecule’s structure from the native ligand, most RMSD for RLDOCK scores were high due to redocking poses of the ligand located well outside the linezolid binding site. This failure is due to the vast search space required for blind docking to ribosomal rRNA.

Figure 3 and Figure 4 summarise the re-docking results of eleven small molecule oxazolidinone complexes using five different program programs for the overall and top-performing values, respectively. Overall, only DOCK 6 and Vina were able to successfully sample at least one structural space of the tested crystal structures with AD4, rDock and RLDOCK. AD4 and Vina consistently had RMSD values over the 2.5 Å cut-off (Figure 3c). DOCK 6 was the most consistent program at reduplicating the native pose and had the highest performing scoring. However, while it was the most successful, it still scored only four of the eleven crystal structures (Figure 3c), and only 52% of the top-performing binding poses (Figure 4c) were under the experimentally binding value of 2.5 Å. While these docking programs are valuable tools for ribosomal docking, significant improvements are needed to adapt to the high flexibility of the binding pockets. For the above reasons, DOCK 6 was selected to continue the vs. section of the study.

### 2.2. Selection of Ribosomal Structure for Virtual Screening

The availability of X-ray crystal and cryogenic-electron microscopy (cryo-EM) structures of ribosomes has provided an avenue for structure-based drug design (SBDD). SBDD has recently become a suitable and powerful tool for antibacterial drug discovery, mainly due to significant advances in structural biology around the bacterial ribosome [38]. Given the knowledge that minor structural differences between bacterial species can affect drug binding, for the progress of SBDD, it is imperative to have a high-resolution crystal structure of the ribosome and its subunits from the pathogenic bacterial species [39]. However, due to the limited number of 50S ribosomes that are bacterial targets, the resolutions ranged from 2.7 Å to 3.5 Å, which is not ideal (Appendix A). The quality of the structures is poor, as the density is not fully resolved and does not fully cover the entirety of the ligands. Similar observations for poor electron densities and species variation have been reported for the structures of the erythromycin, azithromycin, telithromycin, and chloramphenicol complexes with ribosomes [40]. However, to our knowledge, there have been no large-scale vs. studies for a range of ribosome targets. Hence, we aimed to investigate structure-based design despite the low resolutions.

The *S. aureus* crystal structure indicates that linezolid is bound at the PTC, blocking the A-tRNA (aminoacyl binding site, or A-site) in an orientation. This interaction was similarly observed in other ribosome linezolid complexes with 50S *Deinococcus radiodurans* (*D. radiodurans*) [41], *Haloarcula marismortui (H. marismortui)* [42], or 70S *E. coli* [43]. However, unlike these other ribosomal complexes, in *S. aureus*, the flexible nucleotide U2585 [44] undergoes significant rotation. It forms a hydrogen bond with the O4 of the linezolid morpholino ring, leading to a non-productive conformation (shape change) of the PTC [39]. This difference highlights the interest in the specific interaction of linezolid in *S. aureus*. Of the eleven crystal structures chosen, one was of the bacteria *S. aureus*, and six were the ribosome of MRSA. While the MRSA structures had slightly higher resolution and better electron density around the ligand than the *S. aureus* structure, mutation or resistant strains of bacteria undergo induced fit mechanisms in the binding pocket [41]. Induced fit indicates that the active site changes conformation to allow a better fit between the active site and the ligand. As the residues that the structure interacts with are critical to its efficacy, such as the flexible nucleotide U2585 [44] with the O4 in the morpholino ring, any change to the binding pocket is not preferred. Additionally, most derivatives had literature MIC values against *S. aureus*. Amongst the other crystal structures, for DOCK 6, 4WFA was the third highest-performing structure (Figure 3). While 3CPW and 3DLL preceded it, these were crystal structures of *H. marismortui*. Additionally, 4WFA was the highest-performing crystal structure of *S. aureus* origin (including MRSA structures). For the above reasons, we investigated the docking of oxazolidinones to the binding pocket of the 50S *S. aureus* crystal structure (4WFA). 

### 2.3. Virtual Screening (VS) of the Oxazolidinone Dataset

We wanted to determine whether slight modifications in the structures of ligands may influence or impact the docking and scoring of the derivative. For this analysis, the vs. and analysis of the derivatives were done against *S. aureus* (PDB:4WFA) using DOCK 6. We grouped each of the 285 derivatives via predetermined specific structure modification groups. The key groupings can be listed as follows: aryl, base, morpholino, tail, or other oxazolidinones. Descriptions and examples of each modification group are displayed in Figure 5. While the tail group was determined as a distinctive modification group, it was always paired with other modifications (such as Aryl-tail, morpholino-tail, or base-tail), so we created different groups to address the various forms of structural modifications adequately. We ranked each performance by the top-performing score or Top 1, in addition to the averaged Top 3 and Top 5 scores. Most of the dataset scored above −60 kcal/mol, while the top performers were outliers and performed below −85 kcal/mol (Appendix A). Only the eleven top-performing derivatives were used for further analysis, displayed in Table 2.

#### 2.3.1. Structural Modification and MIC Activity

Figure 6a presents the preliminary analysis of the average top performance when docked with DOCK 6 and separated into structural modification groups. The dataset comprised 32.3% of derivatives with an aryl modification (92 derivatives), 27.0% with base modifications (77 derivatives), 15.4% with morpholino modifications (44 derivatives), and 25.3% with other oxazolidinones (72 derivatives). Figure 6b demonstrates a further percentage breakdown of each leading modification group and the tail groups. Structures with the base + tail modification show the highest performance (Figure 6a), followed by morpholino + tail, morpholino, aryl, aryl + tail, base, and other oxazolidinones. Base + tail modifications scored an average highest score (Top 1) of −64.1 kcal/mol, while the lowest average classification group, other oxazolidinones, scored 14.4 kcal/mol higher with an average top 1 score of −49.7 kcal/mol. While there are slight differences between each group of structural modifications, there is no significant correlation between the various derivatives, class of structural modifications, and their scoring performance. As some key structural features of linezolid (such as the oxazolidinone core, 3-fluorophenyl ring, and morpholino moiety) contribute to its high efficacy as a drug target, it is interesting to observe that many of the derivatives with these core structures did not show good performance.

We also separated the derivatives by the MIC activity listed in their literature for *S. aureus* (180 out of the total 285 derivatives) to determine whether a change in activity group influenced how well these derivatives performed. The derivatives with MIC values against *S. aureus* were docked against 4WFA and classified as active (0.01–<4 µg/mL) with 99 derivatives, moderately active (≤4–<16 µg/mL) with 28 derivatives, or inactive (>16 µg/mL) with 53 derivatives. All values were compiled into their specific group and then averaged to show accurate comparisons (Figure 6c). The active and moderately active derivates for *S. aureus* show slightly higher performance than the inactive derivatives. While slight differences exist between the activities, these are not significant values to deem any trend or relationship between the in vitro activity and the performance of the in silico docking. As these derivatives have been reported in the literature, a direct correlation is generally expected between those with lower MIC values and better docking performance. Our current analysis did represent the expected trend, but not substantially. There is a slight difference between active (and moderately active) and inactive; however, this was only an approximate 6 kcal/mol for the top 1. This may suggest inaccuracy in either docking programs/scoring functions or, more likely, due to the high flexibility in the binding pockets of RNA.

The derivatives were separated into groups with or without the O4 position of the morpholino ring (replaced with the R group, Table 2: 1 and 3l). Of the 285 derivatives, 96% had modifications at the O4 position of the morpholino ring. This is an interesting insight, as it has been discussed that in *S. aureus,* the flexible nucleotide U2585 [44] undergoes significant rotation and forms a hydrogen bond with the O4 position of the linezolid morpholino ring, leading to a non-productive conformation of the PTC [39]. However, it cannot be solely attributed to the modification of the O4 of the morpholino ring. It has been previously discussed that the poor posing prediction, in consideration of the electron density and the docking program’s ability to score with RNA, resulted in poor scoring.

#### 2.3.2. Additional Structural Analysis

Based on the previous results, it can be assumed that there is no significant relationship between the derivatives and their structural or inhibitory factors besides sharing a similar base structure. However, to confirm our hypothesis, we wanted to conduct further analysis by comparing other structural factors and their potential linear relationship. Alongside linear regression, we used Pearson’s correlation, which is the linear relationship between two continuous variables. This statistic ranges in (−1, 1), where coefficients greater than zero represent a positive trend, and coefficients less than zero represent an inverse relationship. Initially, we took the absolute docking score of the derivatives and compared them against the negative log of MIC (pMIC) values of *S. aureus*. As expected, the linear relationship with the derivatives against their respecting pMIC is small for *S. aureus*, with a slight positive correlation of 0.396 for *S. aureus* (Appendix A). Using the entire dataset, we further explored structural factors such as the number of rotatable bonds (Appendix A), number of atoms (Appendix A), and molecular weight (Appendix A). Appendix A reveals a positive relationship between the absolute values and structural factors. For the number of atoms and molecular weight, removing the derivatives that performed outside the average range (below −80 kcal/mol) improved the correlation (Pearson’s correlation coefficient improved from 0.18 to 0.36 and from 0.18 to 0.41, respectively). A similar trend occurred for the number of rotatable bonds, but to a lesser extent (Pearson’s correlation coefficient improved from 0.05 to 0.20). Both numbers of atoms and molecular weight display a medium correlation strength, while rotatable bonds and pMIC all display a weak correlation strength. The number of atoms and molecular weight depends on the size of the molecule, so it was expected that these factors might influence how they perform. These results confirm the limited correlation between derivatives factors and their docking performance. However, these findings may be somewhat limited due to the small sample size of the dataset, so it is suggested to incorporate a more extensive dataset in the future.

#### 2.3.3. Top-Performing Derivatives and Their Interactions

We deemed the top-performing values with significant scoring compared to most of the dataset. Out of the 285 derivatives, there were eleven structures determined to be “top-performing” for *S. aureus* (4WFA), those being: **1**, **3l**, **29f, 30d**, **45a**–**45f**, and **67** (Table 2). The derivatives are listed with the MIC values reported in their respective literature for both *S. aureus*. The top 3 poses are superimposed against the native linezolid in Appendix A. Appendix A displays that out of the 11 top-performing derivatives, **3l**, **29f**, and **67** were found to dock within the binding site in a similar orientation and residue interactions as linezolid. Similar to linezolid, which interacts with residues G2088, A2478 U2533 and U2612, derivatives **3l** and **29f** both interact with residues U2533, U2612, and A2478, while derivative **67** interacts with residues G2088 and A2478. Appendix A demonstrates each **3l**, **29f**, and **67** residue interaction in the binding pocket alongside the linezolid. 

**Table 2 antibiotics-12-00463-t002:** List of derivatives determined to be the “top-performing” derivatives, with their similar base structure indicated. The structural differences are indicated by the R, X1. and X2 columns and the docking score and MIC for *S. aureus*.

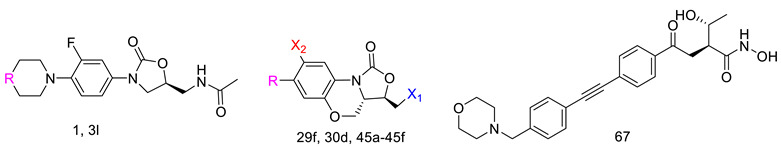
Derivative	R Group	X1 Group	X2 Group	Docking Score (kcal/mol)	*S. aureus* MIC (µg/mL)
**1** [45]	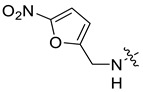	-	-	−94.578	1
**3l** [46]	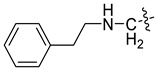	-	-	−90.462	2
**29f** [10]	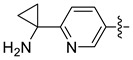	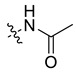	H	−96.613	0.25
**30d** [47]	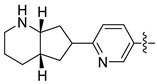	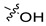	H	−101.091	-
**45a** [48]	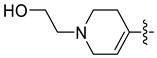	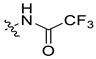	F	−85.279	-
**45b** [48]	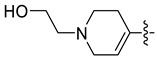	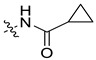	F	−89.955	1.916
**45c** [48]	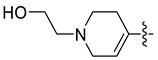	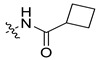	F	−93.988	0.473
**45d** [48]	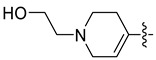	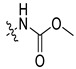	F	−91.024	1.104
**45e** [48]	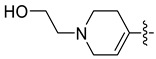	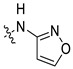	F	−101.503	2.223
**45f** [48]	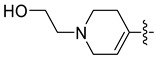	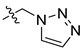	F	−99.681	0.98
**67** [49]	-	-	-	−92.438	-

#### 2.3.4. Scoring Functions

AnnapuRNA is an ML-based scoring function in which the statistical data observed in the experimental RNA–ligand complex are trained to discriminate native structures from decoys [50]. We re-scored the DOCK 6 scoring function to 35 selected numbers of derivatives in the dataset, with five derivatives from each predetermined structural modification. The scoring function removes any potential bias the DOCK 6 scoring function may have had on the results. As shown in Appendix A, AnnapuRNA significantly increased the scoring in all structural modifications. These results demonstrate that combining molecular docking and re-scoring using the AnnapuRNA function is a vital scoring function tool and should be utilised further in future studies.

#### 2.3.5. Random Forest to Systematically Classify Bias in Scoring Functions

A typical representation of chemical structures in these models is the molecular fingerprint, such as Morgan fingerprints. Morgan fingerprints work by assigning each atom of a structure with an identifier, updating said identifier based on its neighbour, removing any duplications, and then folding this list of identifiers into a 2048-bit vector [51]. This examines the local connectivity of each atom in a structure and creates a unique identifier to represent the atom’s local chemical environment [52]. By doing this, we can perform statistical analyses and machine learning techniques on the set of molecules to gain new insights that we could not gain as humans.

We trained a random forest (RF) model to identify potential problem chemical groups that appear to bias a scoring function to over or underpredict the scoring of a compound. RF is a widely used algorithm designed for large datasets with multiple features, as it simplifies by removing outliers and classifying and designating datasets based on relative features classified for the particular algorithm [53]. The model was trained on the Morgan fingerprints of molecules that produced errors in affinity outside the confidence interval in the regression plot, using molecules with known MIC activity against *S. aureus*. The MIC values were converted to pMIC to reduce the skewing of the data. The models in the dataset were classified as over or underpredicted and used in the model’s training. Examples of bit-vectors taken from the molecular fingerprints used to train the models associated with a particular class are shown in Figure 7. As expected, it was difficult to ascertain significant differences in bits between the over- and under-predicting, as most contributing bits were shared across the two classifications due to the high structural similarity between the derivatives in the dataset. However, DOCK 6 tended to underpredict molecules with tail modifications with acetamide, n-methylacetamide or n-ethylacetamide and over-predict molecule derivatives with methylamino bits. However, due to the size of the dataset and high structural similarities between the derivatives, these are not overly contributing factors, and it is suggested to use them in a larger dataset for future analysis to see if these results persist. 

#### 2.3.6. Tuning the Scoring Function/Re-scoring Function

DOCK 6 employs two physics-based scoring functions, termed the Amber score and grid score. The Amber score binding energy ΔG_bind_ is calculated as E_Complex_ – (E_Receptor_ + E_Ligand_), where E_Complex_, E_Receptor_, and E_Ligand_ are MM-GB/SA energies as approximated by the Amber force field. The Amber score enables all or a part of the ligand-receptor complex to be flexible by defining a movable region in the DOCK input. A previous study on RNA–ligand complexes suggested that the performance of this scoring approach is dependent on the number of rotatable bonds in a ligand, requires equilibrated structures [16], and is computationally expensive for a large-scale system such as ribosomal rRNA. In this work, we used the grid scoring of DOCK 6 consisting of intermolecular van der Waals (VDW) and Coulombic energies (scaled by a distance-dependent dielectric) between the ligand and receptor. Using the absolute DOCK 6 grid scoring (to allow a positive trendline), Pearson’s correlation between the predicted and experimented pMIC was 0.485 (Figure 8a).

We tried to tune the re-scoring function using molecular descriptors to improve the poor docking results discussed above. The Gibbs free energy ΔG_bind_ is dependent on the change in enthalpy and entropy, and optimizing these factors can improve the affinity of the ligand. The molecular descriptors such as topological polar surface area (TPSA) and the number of rotatable bonds (nRotB) can contribute to the enthalpic component, hydrophobic effect, and desolvation penalty. The molecular descriptors that considered the overall physical properties of the molecule were LogP, TPSA, nRotB, molecular weight (MolWt), number of H bond donators (HBD), number of H bond acceptors (HBA), LogP, and the number of rings (NumRings). Principal component analysis (PCA) was used to determine relevant molecular descriptors and to find a few common factors controlling all variables to study the relationship between the antimicrobial activity and various parameters for the chosen dataset.

The correlation coefficient between the experimental and predicted pMIC values for the test set compounds (30 compounds) were calculated through multiple linear regression. 

The formula used for the re-scoring function for the test set compounds after regression analysis was:(1)$$\begin{array}{ll} Rescore= & \left(w1\;*\;\mathrm{Docking}\;\mathrm{Score}\right)+\left(w2\;*\;\mathrm{MolWt}\right)+\left(w3\;*\;\mathrm{TPSA}\right)\\ & \;\;+\left(w4\;*\;\mathrm{nRotB}\right)+\left(w5\;*\;\mathrm{HBD}\right)+\left(w6\;*\;\mathrm{HBA}\right)+w7\;\\ & \;\;*\;\mathrm{LogP}+\left(w8\;*\;\mathrm{NumRings}\right)+\;c\; \end{array}$$
where w1 to w8 = weights obtained after regression analysis on the training set compounds;

Docking score = Absolute DOCK 6 docking scores of the test set compounds;

MolWt = Molecular weight of the test set compounds;

TPSA = Topological polar surface area of the test set compounds;

Nrotb = Number of rotatable bonds of the test set compounds;

HBD = Number of H bond donors of the test set compounds;

HBA = Number of H bond acceptors of the test set compounds;

LogP = LogP values of the test set compounds;

NumRing = number of rings of the test set compounds;

c = intercept obtained from regression analysis (−3.02).

This re-score, used to predict pMIC, was a great improvement in the correlation of the docking scoring and known pMIC, going from an *r* or Pearson’s correlation of 0.485 to 0.779 (Figure 8). This improvement in correlation demonstrates that the re-scoring function and descriptors can be used as a predictor of pMIC, even when docking results are not ideal. The full list of the training set with descriptors and pMIC vs. predicted dataset (without the testing set) is shown in Appendix A, respectively.

### 2.4. Limitations of Study

Computational methods for the structure-based docking of small molecules to RNA molecules, such as the 23S ribosome, are not as established as similar methods for protein–ligand docking [22]. Compared with predicting the protein–ligand interaction, modelling the binding interactions between RNA–ligand molecules presents some distinctive challenges [54]. 

rRNA is a large and complex molecule that can adopt a wide range of conformational flexibilities, which can make it challenging to model its binding behaviour accurately as it can fold into multiple stable conformations [55]. The intrinsic flexibility of nucleic acids is often ignored by docking programs. The binding of the ligand to the corresponding RNA can result in either an induced-fit effect or a conformational stabilisation/destabilisation of the complex tertiary structure [56]. Compared with protein–ligand binding, ligand binding sites on RNA can be shallower, highly polar, solvated, and conformationally flexible, adding further complexity when predicting RNA–ligand interactions.

Many molecular docking studies do not consider the associating effects of water molecules and metal ions. Neglecting such solvent effects can cause a significant impact on water or ion-mediated interactions, leading to inaccurate predictions of RNA–ligand interactions. A solution is to use simulations with explicit waters and ions to improve RNA structures [57,58], which allows the results to be sensitive to selecting the critical water molecules. However, achieving complete accuracy can be challenging with this approach, as RNA–ligand interactions are sensitive to the positions and orientations of the water molecules and ions within the cavity space [59]. The prediction of the binding of water molecules and bound metal ions to the RNA before docking, then treating the predicted bound water molecules and ions as part of the receptor for RNA–small molecule docking, is an alternative. Finally, solvation and desolvation effects should also be considered upon ligand binding and water-mediated interactions. While explicit solvation sites within protein–ligand binding pockets can substantially impact affinity and selectivity, similar effects are likely represented in RNA–ligand binding sites as well [60]. 

In addition to these two significant challenges, there are others that limit the performance of RNA docking. The available scoring functions were not trained against rRNA and their ligands; thus, many experimental poses were poorly ranked. There is also a limited number of experimentally determined RNA structures, so the RNA–ligand complex makes knowledge-based approaches less effective for RNA–ligand predictions [54,61]. Another reason is that rRNA has several functional groups and binding sites that can interact with other molecules. This can make it challenging to identify the specific binding site or sites that a drug candidate would need to bind, a prime example being the overlapping binding sites of chloramphenicol and linezolid. Similar to chloramphenicol, linezolid directly clashes with the placement of the aminoacyl moiety of the aa-tRNA [62]. Finally, the binding of drugs to rRNA can be affected by pH, temperature, and the presence of other molecules [63]. 

Overall, molecular docking can be a valuable tool for predicting the binding of small molecules to protein targets, but it may not be as reliable for predicting the binding of drugs to rRNA. These factors can make it difficult to accurately predict the binding behaviour of a drug candidate using molecular docking. More accurate, ensemble-based free energy scoring methods are routinely used to re-score compounds during VS, such as molecular dynamics (MD) simulations [64,65,66], including molecular mechanics with generalized born and surface area (MM/GBSA) [67] and molecular mechanics Poisson–Boltzmann surface area (MM/PBSA) [68] methods. However, it must be noted that these methods are more computationally expensive than docking, usually making them restrictive for extensive-scale compound database screening. Pharmacophore methods are another ligand-based alternative that uses a molecular field points-based similarity method to generate a series of low-energy conformations for each compound. When used in conjunction, docking and pharmacophores can complement each other in revealing critical structural features and could be helpful for the development of highly selective, potent potential drug molecules, especially when the target is either unknown or poor quality (Appendix A) [6,7,8,9]. Additionally, several tools such as NPDock [69], HNADOCK [70] and HDOCK [71] have been developed for protein-RNA/DNA docking and RNA/DNA-DNA/RNA docking. However, structure-based vs. against RNAs are still rare. Future work will focus on validating these docking tools for their applicability against rRNA-small ligand targets, which is crucial because parameters cannot easily be transferred, and as discussed extensively above, nucleic acids hold their own unique challenges.

In summary, for the reasons explained above, these vs. results must be interpreted cautiously and can only guide docking-based virtual screens of oxazolidinone targets.

## 3. Materials and Methods

### 3.1. System Selection

We obtained available crystal structures from the RCSB Protein Data Bank (PDB), only selecting those which were 50S ribosomal structures that contained oxazolidinones as the ligand. Eleven 50S ribosome PDBs were selected from the following organisms: *D. radiodurans* (3DLL) [41], *E. coli* (6QUL) [72], *H. marismortui* (3CPW and 3CXC) [42,73], *S. aureus* (4WFA) [39], and MRSA (6DDD and 6DDG, 6WQN, 6WQQ, 6WRS, and 6WRU) [59,74]. To compare the performance of selected docking programs for use with oxazolidinone and linezolid derivates, we performed docking using AD4 [15], Vina [12], DOCK 6 [16], rDock [19], and RLDOCK [20] with the systems described above. We selected programs due to their commercial availability preference and adaptability to RNA and ribosomal targets.

### 3.2. Pocket Location, RNA–Ligand Preparation, and Docking Protocols for Native Ligands

USCF Chimera (Version 1.15) [37] and BIOVIA Discovery Studio Visualizer (Version 21.1) [8] were used to visualise the PDB structures and for docking preparations. The proteins, ligands, and additional molecules were removed from the structure so that the RNA structure was isolated and prepared for docking. The process was repeated, isolating the native ligand instead of the RNA.

The native pocket location for each crystal structure was defined in Chimera using the Axes/Plane/Centroids method, specifically “Define Centroid”, while the ligand was selected. A 20 × 20 × 20 Å box (v = 8000 Å3) was created around this centroid site to allow the ligand to rotate and flex while not leaving the intended active site. The ligand and RNA files were prepared using Dock prep with hydrogens and charges added in the process, selecting starting residues to be calculated by AMBER ff14SB and other residues to be calculated by AMI-BCC.

### 3.3. Molecular Docking

#### 3.3.1. Ligand Docking with AutoDock Vina (Version 1.2.0)

Parallelized Open Babel & AutoDock suite Pipeline or POAP [75] was used to run Vina and AD4. POAP uses a GNU Parallel to connect the OpenBabel and AutoDock/Vina packages to process ligand preparation, receptor preparation, docking tasks, and result processing via a shell command line. Vina was executed through the POAP pipeline. The exhaustiveness was set to 100, the number of CPUs was left at 8, and the number of generated ligand complexes after vs. was set to 10. 

#### 3.3.2. Ligand Docking with AutoDock4 (Version 4.2.6)

AD4, as described above, was executed through the POAP pipeline. The number of parallel jobs was left at the default value of 24, and the number of generated ligand complexes after vs. was set to 10.

#### 3.3.3. Ligand Docking with DOCK (Version 6.9)

The molecular surface of the crystal structures was generated using the dot molecular surfaces (DMS) tool from Chimera. The DMS calculation commonly fails with large multi-chain structures such as RNA, so RNA within 20 Å of the ligand was selected to ensure the calculation succeeded. The sampling algorithm anchor-and-grow was utlised. The sphgen program generated spheres for the molecular surface, and the grid box and grid were created by Showbox and grid programs, respectively, with a grid spacing of 0.4 Å.

#### 3.3.4. Ligand Docking with rDock

BIOVIA Discovery Studio [8] was used to apply the CHARMm forcefield for all ligands. The reference ligand method in rDOCK was used to define the binding site, and the cavity mapping parameters were optimised for each ligand. The docking results were evaluated using the default scoring function SF3 (without desolvation terms) and SF5 (with desolvation terms). The remaining parameters were left as their default values, and the radius of the large sphere was not defined.

#### 3.3.5. Ligand Docking with RLDOCK

RLDOCK uses global docking for the RNA receptor, so the binding site was not specified. The number of simulation threads was set to 32, and the number of output poses after clustering was set to 100. 

#### 3.3.6. Normalisation and Re-scoring of Redocking

The scoring functions of AD4, Vina, DOCK 6, rDock, and RLDOCK programs use a variety of internal scoring functions (Table 1), resulting in different ranges or scoring prior to mean calculation and other statistical operations on the docking scores. Data normalization was performed using the formula below to bring all the scores in a notionally common scale from 0 to 1:(2)$$Normalised\;score\;=1-\frac{\left(x-\min\right)}{\left(\max-\min\right)}$$
where *x* represents the corresponding score, max represents the maximum score, and min represents the minimum score of the dataset.

### 3.4. Virtual Screening (VS) with Oxazolidinone Derivatives

Virtual screening was performed with a dataset comprising 285 oxazolidinone and linezolid derivatives, compiled from multiple studies in Zhao et al.’s review on oxazolidinone scaffolds [6]. The derivative structures were taken from the review paper, manually drawn, and placed in a spreadsheet using ChemDraw. Then, these were converted into the simplified molecular-input line-entry system (SMILES) with their MIC against *S. aureus* noted, retaining their chirality and stereochemistry. The MIC values were also converted to pMIC for additional analysis using the equation below (See Figure 7 and Appendix A).
pMIC = −log_10_[MIC](3)

Ligands for virtual screening were prepared as stated above in DOCK 6. Unlike AD4, Vina, RLDOCK, and rDOCK, DOCK 6 defines its binding site using spherical data. Spheres within 8.5 Å of the native ligand binding site allowed the derivatives to move and flex but were also small enough to encapsulate the native binding site. Structural modification and MIC activity analysed each derivative in two distinctive categories. Each derivative was placed into a group according to its modification to the linezolid structure, divided into the sub-categories shown in Figure 5: morpholino, aryl, base, tail and other oxazolidinone modifications. Some structures have multiple modifications, such as aryl-tail, morpholino-tail or base-tail, and these modifications were split into separate groups in the analysis. We separated these structures into these groups to determine whether a change in a structural group or MIC activity would positively or negatively alter the performance or score of each derivative. Derivatives with either *S. aureus* values were separated according to the following three different levels of activity: active (0.01–<4 µg/mL), moderately active (≤4–<16 µg/mL), or inactive (>16 µg/mL). The crystal structure 4WFA was used for the virtual screening of all derivatives. 

### 3.5. Re-scoring of Docking Results with AnnapuRNA

AnnapuRNA, a knowledge-based scoring method [25], was used as a re-scoring function on a sample of the vs. oxazolidinone ligands. AnnapuRNA uses a coarse-grained (simplified) representation of both interacting partners. For RNA, only the canonical A, G, C, and U residues were considered. For the ligand poses, they applied the concept of modeling pharmacophoric features by Gaussian 3D volumes instead of the more common point or sphere representations [76]. The RNA crystal structure and DOCK 6 docked ligand poses were scored using AnnapuRNA’s k-nearest neighbors or “kNN modern” method [25]. The results were then split into their modification groups for comparative analysis.

### 3.6. Statistical Analysis and Classifier

Pearson’s correlation calculations were performed using the functions available in the SciPy statistics library in Python 3.6. Several random forest classifiers were generated from the data and individually examined, determining whether the docking program over or underpredicted the affinity of the derivative. This was then compared to the linear regression between the pMIC value and the absolute value of the docking score. The random forest classifier model was built using the Scikit-learn library in Python 3.6 and trained on the Morgan fingerprint bit vector (radius = 2, encoding = 2048 bits), which was generated using RDKit (v. 2021.09.4) for each of the derivatives from the Zhao et al. dataset [6] that had *S. aureus* pMIC values.

### 3.7. Principal Component Analysis (PCA) and Re-scoring Function

Molecular descriptors were calculated using RDKit (v. 2021.09.4), including molecular weight (MolWt), topological polar surface area (TPSA), number of rotatable bonds (nRotB), number of H bond donators (HBD), number of H bond acceptors (HBA), LogP, and the number of rings (NumRings), as well as the DOCK 6 scoring of 4WFA. Regression analysis was performed on the training set compounds with S. aureus pMIC values. The training set comprised 30 randomly selected derivatives, 10 from the active group, 10 from the moderately active group, and 10 from the inactive group, to provide an even spread of data (Appendix A). PCA or principal component analysis was used through RDKit and Sklearn python packages, unitizing a k-means clustering of 3. The efficiency of the re-scoring equation was determined using r or Pearson’s coefficient.

## 4. Conclusions

In this study, we evaluated the performance of five RNA adaptable docking programs in reproducing the crystallographic pose of ligands located in the native binding site of each target, which in this case, was oxazolidinones. Here, we studied several leading docking programs or algorithms, namely, DOCK 6, AutoDock 4, AutoDock Vina, rDock and RLDOCK, and evaluated their ability to correctly reproduce and score the crystal structure ligand configuration for eleven oxazolidinone crystal ribosomal structures. However, we caution readers about the use of these ribosomes in future studies due to the poor electron density and quality of said structures. Through the validations, we found AutoDock, AutoDock Vina, and DOCK 6 tended to perform better when redocking the native ligands compared to rDock and RLDOCK. RLDOCK had the highest success rate for docking under ≤2.5 Å. However, RLDOCK was excluded due to the program’s global docking-only function. DOCK 6 tended to have a much greater scoring than other programs, with the lowest median RMSD and the second highest success rate for docking under ≤2.5 Å.

We reported that the structural modifications, factors, and type of MIC activity of a 285-oxazolidinone dataset had minimal to no influence on how the linezolid derivatives performed when docked. We found ten derivatives that scored within the “top-performing” range, with 3 of those binding in the native binding site. However, due to the complexity of rRNA, such as the high flexibility of the binding pocket, which resulted in poor pose prediction and poor posing of the derivatives, results must be interpreted with caution and can only guide docking-based virtual screens of oxazolidinone targets. Morgan fingerprint analysis determined that DOCK 6 tended to over-predict derivatives with acetamide-containing tail modifications and over-predict those with methylamino bits. However, due to the high similarity between each derivative and the size of the dataset, these results may not be major contributing factors. We also suggested that scoring function, re-scoring equation, and pharmacophore methods can be used in conjunction with docking. The re-scoring equation method showed significant improvement in the predictions of the training set. Additionally, further RNA–ligand studies should consider water, pH, and metal ions, as these factors affect the stability of the binding site. Additional methods, such as MD, should also be considered. In conclusion, the current results suggest that one should perform further docking studies and vs. campaigns of oxazolidinones, adapt additional analysis methods, and utilise a more extensive dataset.

## Figures and Tables

**Figure 1 antibiotics-12-00463-f001:**
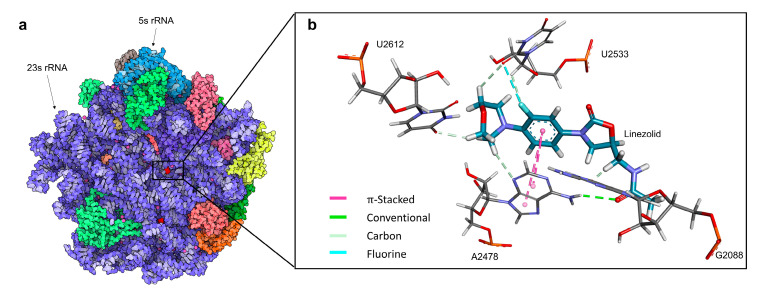
The *S. aureus* (strain NCTC832) 50S ribosome crystal structure (PDB ID 4WFA) with 50S ribosomal proteins (not all proteins are shown for clarity) and 5s and 23s rRNA ribosomes coloured and labelled with (**a**) rotated side view of the crystal structure with the linezolid binding site displayed. (**b**) Close-up view of the linezolid at the ribosomal binding site. Linezolid is shown in aqua blue, while the residues that interact in the active site are coloured by element and labelled with their corresponding residue name, with *E. coli* rRNA numbering. The interactions between the *S. aureus* residues and the ligand in the binding pocket are shown via the striped line, coloured by the following interactions: π-stacked (pink), conventional (bright green), carbon (light green), and fluorine (blue). Images were created using Protein Imager [4] and BIOVIA Discovery Studio Visualizer [8].

**Figure 2 antibiotics-12-00463-f002:**
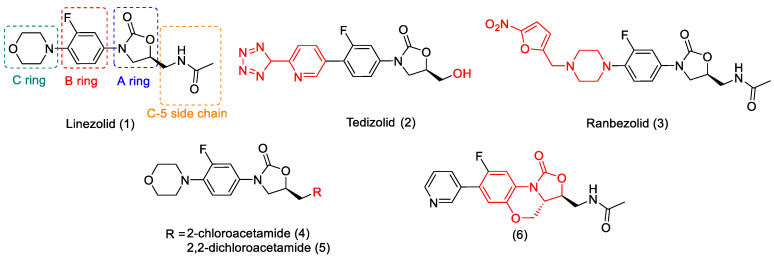
Oxazolidinones on the market are linezolid (1) and tedizolid (2), with the A ring, B ring, C ring and C-5 side chain indicated on linezolid. Examples of modifications to the linezolid scaffold include morpholine modification (3), C-5 side-chain modifications (4) and (5), and oxazolidinone modification (6) from the oxazolidinone dataset. The image was created using ChemDraw.

**Figure 3 antibiotics-12-00463-f003:**
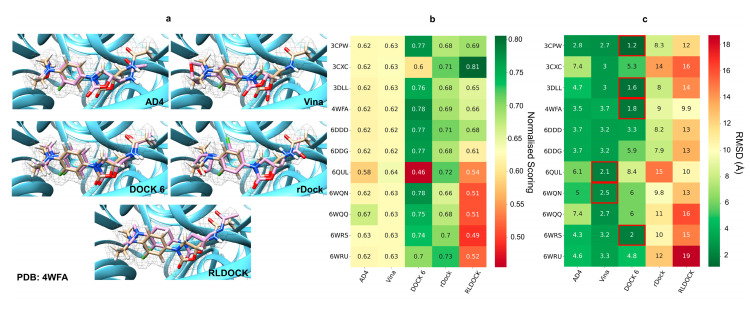
Comparison of the redocking performance for the linezolid. (**a**) Examples of results produced when redocking linezolid to its native crystal structure (PDB: 4WFA). The native ligand pose is beige, the selected redocked ligand is pink, and the 2 Å mesh of the electron density map is grey. (**b**) The average scoring for each program is displayed in a heatmap using normalized scoring values, showing all linezolid ligand crystal structures. (**c**) The average RMSD for each program is displayed in a heatmap, showing all linezolid ligand crystal structures. The crystal structures that scored at or below 2.5 Å are indicated by red squares. Figure 3a was visualised through UCSF Chimera (v 1.15) [37].

**Figure 4 antibiotics-12-00463-f004:**
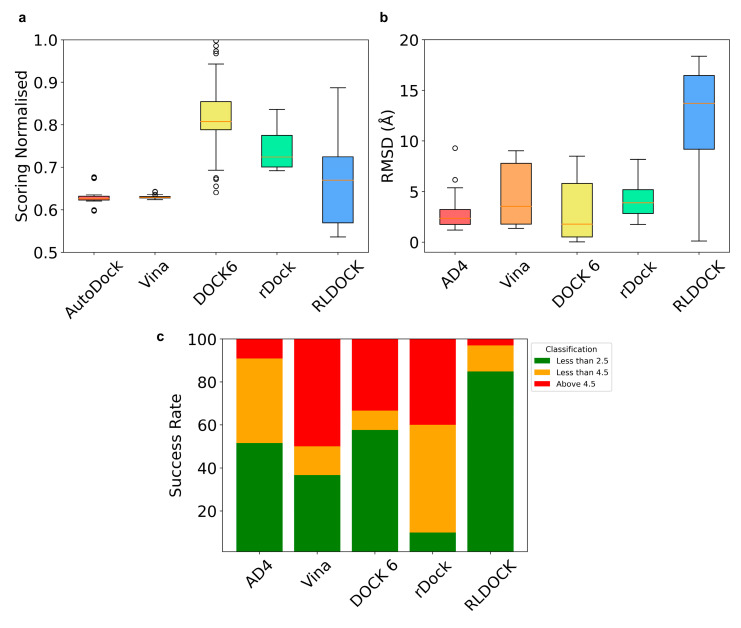
The five selected docking programs used to validate crystal structures using the top-performing values from each crystal structure. Each docking program was ranked by scoring (**a**) and RMSD (**b**). Each program’s overall success rate was used to calculate the average using the normalized top-scoring poses of each program. Both graphs display the minimum and maximum values, with the median values indicated with the orange line and the circles being the outliers. (**c**) A stacked bar chart demonstrating each selected program’s success rate is shown. The green, yellow, and red areas correspond to the percentage of models that achieve an RMSD of ≤2.5, ≤4.5, and >4.5 Å, respectively.

**Figure 5 antibiotics-12-00463-f005:**
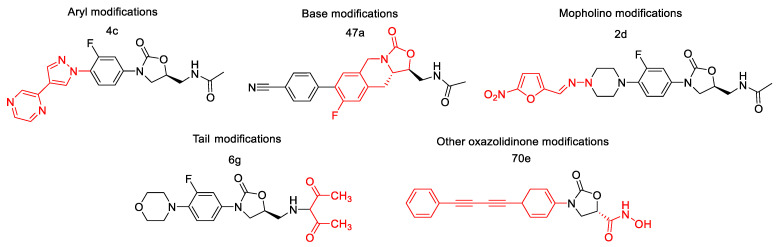
The following structures show the modification to the oxazolidinone derivatives based on the linezolid structure in red. They are split into the following categories: morpholino modifications (**2d**), aryl modifications (**4c**), base modifications (**47a**), tail modifications (**6g**), and other oxazolidinone modifications (**70e**). Note that some structures have multiple areas of modification. This image is to detail an example of what these modifications may be.

**Figure 6 antibiotics-12-00463-f006:**
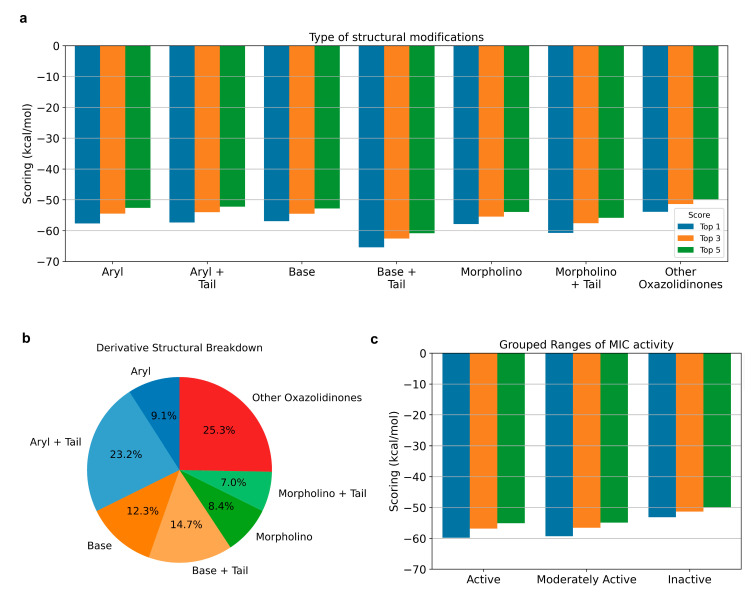
Docking performance of DOCK 6 towards key structural modification of linezolid and correlation with the MIC against *S. aureus*. (**a**) The average performance of structural modifications when docked with DOCK 6. There are some slight differences between each modification, but nothing of significance. (**b**) Percentage breakdown of the number of structural modifications. (**c**) Some derivatives had activity against *S. aureus.* Similar to the structural modifications, MIC activities had no significant difference.

**Figure 7 antibiotics-12-00463-f007:**
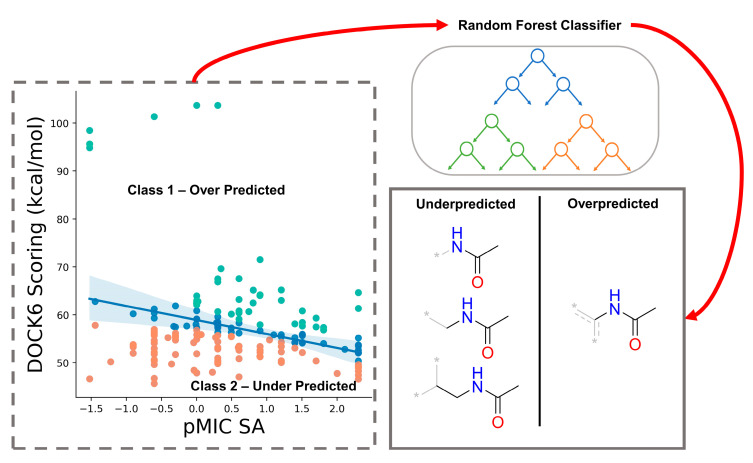
A brief overview of the random forest classifier. The classifier is trained on each derivative’s Morgan fingerprint bit vectors to identify sections of the structure that may contribute to over- or under-predictions by the scoring function. Over-predicted derivatives are indicated by green and under-predicted by orange. The top-performing derivatives are outliers in the −90 kcal/mol to −100 kcal/mol range.

**Figure 8 antibiotics-12-00463-f008:**
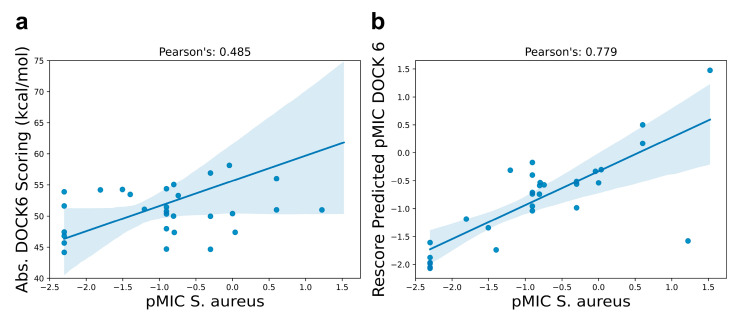
Scatter plots showing coefficient of Pearson’s correlation (r) between the experimental pMIC and predicted pMIC for test set compounds. (**a**) Absolute DOCK 6 score. (**b**) DOCK 6 re-score.

**Table 1 antibiotics-12-00463-t001:** Summary of the target, scoring functions, and search algorithms of the programs tested in the study.

Program	Target	Scoring Function	Search Algorithm
AutoDock 4	Protein	Physics-based + empirical	Lamarckian genetic algorithm
AutoDock Vina	Protein	Physics-based + empirical	Monte Carlo and quasi-Newton
DOCK 6	RNA	Physics-based + force field	Incremental construction
rDOCK	Protein/RNA	Physics-based + empirical	Genetic algorithm, Monte Carlo and simplex minimization
RLDOCK	RNA	Physics-based + empirical	Multiconformer docking

## Data Availability

Molecular dockings were performed with AutoDock 4, AutoDock Vina, DOCK 6, rDock, and RLDOCK, using the cited versions, all open sources. Pharmacophore performed Cresset Flare was also using cited versions, which can be accessed through a commercial license. Scoring via AnnapuRNA is available through GitHub via the cited version. All visualisation sources used for the preparation of Figures and Appendix A are listed in the captions of the figures, with python visualisation toolkits used if not stated otherwise. All prepared oxazolidinone scaffolds from the dataset are available at https://github.com/gandhi-group-biomolecular/docking-oxazolidinone-dataset, accessed on 23 February 2023.

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
