# Peer review of "Comparative Assessment of Docking Programs for Docking and Virtual Screening of Ribosomal Oxazolidinone Antibacterial Agents"

_antibiotics, 2023, doi:10.3390/antibiotics12030463_

Round 1

Reviewer 1 Report

The manuscript entitled “Validation of nucleic acid molecular docking programs for virtual screening of oxazolidinone antibacterial agents (antibiotics-2193858)” reports docking of oxazolidinones by five docking programs. The manuscript cannot be recommended for publication in antibiotics because of following reasons:

1.      Title of the manuscript states “Validation of nucleic acid molecular docking programs” but programs included in the study are not specifically used for nucleic acid docking; rather those programs are frequently used for protein-ligand or protein-protein dockings and have been validated several times before. Therefore, the title itself is misleading and lacks rationality, novelty as well as applicability in the drug discovery projects. Ideally, the study should include a number of specific programs which deals with ligand–RNA binding predictions such as RLDOCK.

2.      There are several validation and benchmarking studies in the literature and hence where is the novelty in this study?

3.      The presentation is very poor and figures must be improved. In particular, Figure 9.

4.      The docked poses must be further analyzed by molecular dynamics simulation in order to ascertain conformational and thermodynamic stability of the complexes.

Author Response

Reviewer 1

Comments and Suggestions for Authors

The manuscript entitled “Validation of nucleic acid molecular docking programs for virtual screening of oxazolidinone antibacterial agents (antibiotics-2193858)” reports docking of oxazolidinones by five docking programs. The manuscript cannot be recommended for publication in antibiotics because of following reasons:

  1. Title of the manuscript states “Validation of nucleic acid molecular docking programs” but programs included in the study are not specifically used for nucleic acid docking; rather those programs are frequently used for protein-ligand or protein-protein dockings and have been validated several times before. Therefore, the title itself is misleading and lacks rationality, novelty as well as applicability in the drug discovery projects. Ideally, the study should include a number of specific programs which deals with ligand–RNA binding predictions such as RLDOCK.

Response 1

Thank you for your comment. We have adjusted the title of the manuscript from “Validation of nucleic acid molecular docking programs for virtual screening of oxazolidinone antibacterial agents” to “Comparative assessment of docking programs for docking and virtual screening of oxazolidinone antibacterial agents” to improve the clarity of the study.

Throughout the study, we use docking programs explicitly made for ligand-RNA binding predictions, such as RLDOCK or those that have since been adapted to deal with RNA interactions, such as AutoDock 4, AutoDock Vina, DOCK 6, and rDOCK (see Table 1). These docking programs, while some classically made for protein ligands, have been used in a multitude of studies conducted for RNA, demonstrated in studies utilising 31 complexes (Detering et al., J. Med. Chem. 2004, 4188–4201), 34 complexes (Chen et al. J Chem Inf Model 2012, 2741–2753), 56 complexes (Ruiz-Carmona et al.  PLoS Comput Biol. 2014), among others (Zhou and Jiang et al. WIREs Comput Mol Sci2022).

  1. There are several validation and benchmarking studies in the literature and hence where is the novelty in this study?

Response 2: Thank you for raising this concern. A review paper conducted Zhou and Jiang demonstrates that these programs also been applied to study RNA, however they did not specifically focus on ribosomal RNA (rRNA), which is the focus of our study. While there are several validation and benchmarking studies in the literature, this is the first validation, benchmarking, and virtual screening study on a large dataset of oxazolidinones on rRNA targets. Additionally, comparative assessment studies improve the understanding of the limitations and the docking programs’ competency.

Ligand-based approaches have been traditionally used for studying this class of compounds, typically demonstrating structural modifications to the oxazolidinone (Zhao et al. J. Med. Chem. 2021, 10557–10580). Docking is typically only carried out on the lead molecule from the series to explain binding mode, but not to understand structure-activity relationships or for rationale design, as it was traditionally thought.

  1. The presentation is very poor and figures must be improved. In particular, Figure 9.

Response 3: Thank you for bringing up this concern. We have since improved the quality and resolution of all figures used in the text.

  1. The docked poses must be further analyzed by molecular dynamics simulation in order to ascertain conformational and thermodynamic stability of the complexes.

Response 4: Thank you for bringing up this concern. This study was designed as a docking validation and virtual screening to a rigid RNA target. Some of the advanced methods, such as Molecular Dynamics (MD) and relative free energy scoring methods like Molecular Mechanics with Generalized Born and surface area (MM/GBSA) and Molecular Mechanics Poisson–Boltzmann surface area (MM/PBS) allow the incorporation of explicit water, metal ions and allow flexibility of both ligand and RNA target which are usually neglected by docking algorithms. However, a previous study has shown that rescoring performance is very much dependent on the equilibration of structures before docking (Lang et al. RNA. 2009, 1219-1230). The 23S rRNA studied in this work is a large-scale system (2904 nucleotides in E. coli) and hence can be sampled only through long-timescale computationally expensive simulations. We only considered a high throughput screening approach in this study, and our future studies aim to investigate these systems using methods such as molecular dynamics simulations. However, these methods were out of scope for the current study.

However, we have since added an additional section using a rescoring function (see 2.3.6 Re-scoring function and 3.7 PCA and rescoring function), which uses DOCK 6 scoring of the derivatives and their molecular descriptors. This fine-tuning of the scoring function allowed to predict its potential pMIC, of oxazolidinone derivatives without considering rRNA factors. We hope the inclusion of this section has addressed your concerns.

Docking is a computational technique to predict the binding of small molecules with drug targets. However, in our work no single docking program or scoring function that are commonly used in literature was suitable for rRNA drug targets, so the selection of a suitable program depends on the specific needs of the target. Tuned scoring functions can improve docking accuracy. We are currently working on machine learning algorithms like support vector machines (SVM) that include descriptors and fingerprints for in silico pMIC predictions and it will be published as a separate work.

Reviewer 2 Report

In this paper, the authors use computational methods to model the binding of oxazolidinone antibacterial agents to the bacterial ribosome. In the first part of the study, the authors compare five different docking programs to assess which program is better. Subsequently, the best-performing docking tool (DOCK 6) was used to evaluate 285 oxazolidinone derivatives against oxazolidinone binding sites in S. aureus ribosome. Finally, they use a ligand-based approach to expand their results. The study concludes that the docking results should be taken with caution, that there are no clear correlations between the activity of these antimicrobial agents and the docking predictions, and that other methods, such as MD or experimental tests, should be used to confirm and improve the docking data.

The study is certainly well-designed, and the manuscript is well-written. However, some parts of the text can be improved, and some calculations can be added to improve the predictions. Furthermore, the conclusion could be enhanced based on further calculations and other considerations.

1.     Keywords: the keyword 'ribosomes' could be added. Conversely, the keyword 'RNA' is irrelevant because there are different types of RNA (mRNA, ncRNA, sRNA, asRNA, tRNA). The authors could delete the keyword 'RNA' and/or replace it with 'rRNA'.

2.     Fig. 1. In figure 1, the authors indicate the binding pocket with a star. This is difficult to see. is it possible to highlight the pocket differently?

3.     In the introduction, the authors move from talking about oxazolidinones (lines 48-106) to the approach of docking, with no apparent connection (lines 113-131). This transition in the text could be improved.

4.     The final part of the introduction talks about the rationale of the study (lines 132-147). However, the authors merely list the methods without specifying the rationale (lines 145-147). The rationale should be better explained, clarifying why they use these strategies to improve or validate docking results.

5.     Section 2.1. The authors introduce the docking software without explaining how they chose them and whether there are differences between the different software algorithms. A small table summarizing the characteristics of the various software might help to understand the text.

6.     In figure 3 panel (b) at the top there is a cut-out text that probably needs to be removed.

7.     Fig. 3. is it possible to better highlight the positive results (>2.5 Å) in Figure 3 panel (b)?

8.     Fig. 4 panel (a). The authors compare the different energy values (Kcal/mol) associated with each docking software. This serves to compare the different programs. However, the docking scoring function even though it reports a binding free energy value should not be understood as real free energy, but only as a function to rank the results of the same software. Therefore, it makes no sense to compare this value of different software. This limitation could be curbed by normalizing the values of one software using the average value (or another value) produced by the same software.

9.     Fig. 4. Complexes generated by docking could also be subjected to other software that calculates free energy. For example, "Prodigy" or other similar software. This could help you compare various software based on Kcal/mol values.

10.  Figure 7 as Figure 3. It doesn't make sense to compare free energy values resulting from different software. As before (comment 9) would it be possible to normalize? Or is it possible to use Prodigy or another similar tool to compare the Kcal/mol? It is not clear whether AnnapuRNA only calculates other energy values or whether it refines the docking complexes and then calculates the energy values.

11.  The authors conclude that the docking results are not reliable. However, there are methods for refining the docking data. Why were these tools not used?

12.  In the supplementary file, the name 'S. aureus' is not italicized.

Author Response

Reviewer 2

Comments and Suggestions for Authors

In this paper, the authors use computational methods to model the binding of oxazolidinone antibacterial agents to the bacterial ribosome. In the first part of the study, the authors compare five different docking programs to assess which program is better. Subsequently, the best-performing docking tool (DOCK 6) was used to evaluate 285 oxazolidinone derivatives against oxazolidinone binding sites in S. aureus ribosome. Finally, they use a ligand-based approach to expand their results. The study concludes that the docking results should be taken with caution, that there are no clear correlations between the activity of these antimicrobial agents and the docking predictions, and that other methods, such as MD or experimental tests, should be used to confirm and improve the docking data.

The study is certainly well-designed, and the manuscript is well-written. However, some parts of the text can be improved, and some calculations can be added to improve the predictions. Furthermore, the conclusion could be enhanced based on further calculations and other considerations.

  1. Keywords: the keyword 'ribosomes' could be added. Conversely, the keyword 'RNA' is irrelevant because there are different types of RNA (mRNA, ncRNA, sRNA, asRNA, tRNA). The authors could delete the keyword 'RNA' and/or replace it with 'rRNA'.

Response 1: As per the reviewer's suggestion, we have adjusted the keywords to be more specific, changing RNA to rRNA.

  1. Fig. 1. In figure 1, the authors indicate the binding pocket with a star. This is difficult to see. is it possible to highlight the pocket differently?

Response 2: The clarity of the binding site in Figure 1 has been improved. 

  1. In the introduction, the authors move from talking about oxazolidinones (lines 48-106) to the approach of docking, with no apparent connection (lines 113-131). This transition in the text could be improved.

Response 3: Thank you for raising this concern. We have modified the sentences between paragraphs (lines 117 to 121) to help improve the connection.

  1. The final part of the introduction talks about the rationale of the study (lines 132-147). However, the authors merely list the methods without specifying the rationale (lines 145-147). The rationale should be better explained, clarifying why they use these strategies to improve or validate docking results.

Response 4: We have since added additional clarification between lines 159 and line 170.  

  1. Section 2.1. The authors introduce the docking software without explaining how they chose them and whether there are differences between the different software algorithms. A small table summarizing the characteristics of the various software might help to understand the text.

Response 5: Thank you for the suggestion. We have added a Table (Table 1) summarising the characteristics (i.e. scoring functions and search algorithms) of the various software used.

  1. In figure 3 panel (b) at the top there is a cut-out text that probably needs to be removed.

Response 6: The quality of the images has since been improved, and the cut-out text has been removed.

  1. 3. is it possible to better highlight the positive results (>2.5 Å) in Figure 3 panel (b)?

Response 7: We have since highlighted each positive result (>2.5 Å) with a red box to improve the clarity of the results.

  1. Fig. 4 panel (a). The authors compare the different energy values (Kcal/mol) associated with each docking software. This serves to compare the different programs. However, the docking scoring function even though it reports a binding free energy value should not be understood as real free energy, but only as a function to rank the results of the same software. Therefore, it makes no sense to compare this value of different software. This limitation could be curbed by normalizing the values of one software using the average value (or another value) produced by the same software.
  2. Fig. 4. Complexes generated by docking could also be subjected to other software that calculates free energy. For example, "Prodigy" or other similar software. This could help you compare various software based on Kcal/mol values.
  3. Figure 7 as Figure 3. It doesn't make sense to compare free energy values resulting from different software. As before (comment 9) would it be possible to normalize? Or is it possible to use Prodigy or another similar tool to compare the Kcal/mol? It is not clear whether AnnapuRNA only calculates other energy values or whether it refines the docking complexes and then calculates the energy values.

Response 8 - 10: Thank you to the reviewer for bringing up this concern. However, Prodigy only allows the calculation protein-protein and protein-ligand interactions and does not allow the incorporation of nucleic acids. We have since normalised the scoring for all docking programs, to make an accurate comparison. We have since added a section in the methods detailing the methods used to normalise the data, under the section in Methods titled “3.3.6 Normalisation of docking scoring functions”. The heatmap for scoring suggested by another reviewer (updated Figure 3b) contains these normalised values, and Figure 4a and the corresponding text have been updated.

AnnapuRNA (Stefaniak et al. PLoS Comput Biol. 2021) is a knowledge-based scoring function designed for predicting interactions of RNA with small-molecule ligands. AnnapuRNA is used as a rescoring tool, so it only calculates energy values/rescoring the docking scoring. This information is in the methods section, line 887 to line 891.

  1. The authors conclude that the docking results are not reliable. However, there are methods for refining the docking data. Why were these tools not used?

Response 11 Thank you for bringing up this concern. This study was designed as a docking validation and virtual screening to a rigid RNA target. Some of the advanced methods, such as Molecular Dynamics (MD) and relative free energy scoring methods like Molecular Mechanics with Generalized Born and surface area (MM/GBSA) and Molecular Mechanics Poisson–Boltzmann surface area (MM/PBS) allow the incorporation of explicit water, metal ions and allow flexibility of both ligand and RNA target which are usually neglected by docking algorithms. However, a previous study has shown that rescoring performance is very much dependent on the equilibration of structures before docking (Lang et al. RNA. 2009, 1219-1230). The 23S rRNA studied in this work is a large-scale system (2904 nucleotides in E. coli) and hence can be sampled only through long-timescale computationally expensive simulations. We only considered a high throughput screening approach in this study, and our future studies aim to investigate these systems using methods such as molecular dynamics simulations. However, these methods were out of scope for the current study.

However, we have since added an additional section using a rescoring function (see 2.3.6 Re-scoring function and 3.7 PCA and rescoring function), which uses DOCK 6 scoring of the derivatives and their molecular descriptors. This fine-tuning of the scoring function allowed to predict its potential pMIC, of oxazolidinone derivatives without considering rRNA factors. We hope the inclusion of this section has addressed your concerns.

Docking is a computational technique to predict the binding of small molecules with drug targets. However, in our work no single docking program or scoring function that are commonly used in literature was suitable for rRNA drug targets, so the selection of a suitable program depends on the specific needs of the target. Tuned scoring functions can improve docking accuracy. We are currently working on machine learning algorithms like support vector machines (SVM) that include descriptors and fingerprints for in silico pMIC predictions and it will be published as a separate work.

  1. In the supplementary file, the name 'S. aureus' is not italicized.

Response 12: Thank you for pointing this out. The format of 'S. aureus' in the supplementary file has been adjusted.

Reviewer 3 Report

In this study Buckley et al perform a systematic analysis of molecular docking of oxazolidinone within ribosome structures. By comparing the docking results of different docking programs with experimental crystallographic structures, they rank the different programs, noting that they are all limited in terms of replicating the crystal structure results. The authors further test docking of a number of oxazolidinone derivatives using a single program (DOCK6), concluding that the program cannot unravel what makes a compound more or less active. The authors also test alternative ways to calculate the scoring functions of the docking, including using a different program (AnnapuRNA) and random forest approaches. Overall, the manuscript mostly highlights the limitations of molecular docking of small molecules, suggesting some potential avenues to pursue to improve this.

The manuscript is overall well written, and represents a thorough investigation. However, I don’t understand how Fig 3b relates to Figs 4b and 4c. I believe that 3b is showing the average RMSD between the docking and the actual crystal structure. Therefore I think that Fig 3b represents the data points that went into Fig 4b and 4c. However, line 178 and Fig 3b state that for AD4 “none of the crystal structures obtaining a value of less than 2.5 Å”; but then line 181 and Figs 4b & 4c suggest that “52% of the top-performing values were successful (being below 2.5 Å)”. The same issue applies to the performance description for the other programs used in the study. Therefore the authors need to explain in a clearer way what is shown in Fig 3b vs Figs 4b & 4c, as this is one of the key results of the paper.

Minor comments:

-       Line 169: should read “The docked poses were considered successful” (instead of will be successful).

-       It would be useful to show the datapoints for Fig 4a (the docking scoring), e.g. by showing a heatmap like Fig 3b for RMSD.

-       Figure 6 and related text: the authors should indicate the number of compounds tested for each type of structural modification and for the different MIC activity groups.

-       Line 273: I suggest adding the number of derivatives tested; otherwise this is not mentioned until line 330. E.g. “We grouped each of the 285 derivatives”

-       Line 373: “3l, 29f, and 67 were successfully found to dock” how are the authors defining a successful docking? Looking at Sup Fig 4 the docking would be above the 2.5 Å RMSD they used in the initial screen.

-       Line 481: There is a repeated sentence “Many molecular docking studies do not consider the associating effects of water molecules and metal ions”.

Author Response

Reviewer 3

Comments and Suggestions for Authors

In this study Buckley et al perform a systematic analysis of molecular docking of oxazolidinone within ribosome structures. By comparing the docking results of different docking programs with experimental crystallographic structures, they rank the different programs, noting that they are all limited in terms of replicating the crystal structure results. The authors further test docking of a number of oxazolidinone derivatives using a single program (DOCK6), concluding that the program cannot unravel what makes a compound more or less active. The authors also test alternative ways to calculate the scoring functions of the docking, including using a different program (AnnapuRNA) and random forest approaches. Overall, the manuscript mostly highlights the limitations of molecular docking of small molecules, suggesting some potential avenues to pursue to improve this.

The manuscript is overall well written, and represents a thorough investigation. However, I don’t understand how Fig 3b relates to Figs 4b and 4c. I believe that 3b is showing the average RMSD between the docking and the actual crystal structure. Therefore I think that Fig 3b represents the data points that went into Fig 4b and 4c. However, line 178 and Fig 3b state that for AD4 “none of the crystal structures obtaining a value of less than 2.5 Å”; but then line 181 and Figs 4b & 4c suggest that “52% of the top-performing values were successful (being below 2.5 Å)”. The same issue applies to the performance description for the other programs used in the study. Therefore the authors need to explain in a clearer way what is shown in Fig 3b vs Figs 4b & 4c, as this is one of the key results of the paper.

Response: Thank you for your kind feedback; it is greatly appreciated. An additional heat map was included for normalised scoring as shown in Figure 3b, while the existing RMSD heatmap is now Figure 3c. Figure 3b and 3c represent the overall average scoring/RMSD for each crystal structure and docking program. In contrast, Figure 4a and 4b represent the top-performing scoring/RMSD values for the docking programs only. We have added an additional paragraph to clarify what both images represent and modified text from line 201 to line 269 to improve the clarity. We have also place the Figure 3 discussion and Figure 4 discussion in their own individual paragraphs, to avoid jumping between the figures.

Minor comments:

  1. Line 169: should read “The docked poses were considered successful” (instead of will be successful).

Response 1: Thank you for highlighting this error. We have adjusted the sentence from “will be successful” to “were considered successful”.

  1. It would be useful to show the datapoints for Fig 4a (the docking scoring), e.g. by showing a heatmap like Fig 3b for RMSD.

Response 2: Thank you for the suggestion. A heat map has been added to Figure 3, with scoring being figure 3b and RMSD being 3c. It was suggested by another reviewer to normalise these values, so the heatmap contains these values and Figure 4a, Supplementary Figure 3 (and any corresponding text) have since been updated. The figure numbering has also been adjusted in the text.

  1. Figure 6 and related text: the authors should indicate the number of compounds tested for each type of structural modification and for the different MIC activity groups.

Response 3: Thank you for the suggestion. We have added a specific number of modifications in each group in the text, from lines 371 to 373.

  1. Line 273: I suggest adding the number of derivatives tested; otherwise this is not mentioned until line 330. E.g. “We grouped each of the 285 derivatives”

Response 4: Thank you for the suggestion. To improve clarity, we have mentioned the number of derivatives at an earlier point in the text where the dataset is first mentioned, at line 156 .

  1. Line 373: “3l, 29f, and 67 were successfully found to dock” how are the authors defining a successful docking? Looking at Sup Fig 4 the docking would be above the 2.5 Å RMSD they used in the initial screen.

Response 5: Successfulness of the docking was deemed via the positioning of the docked ligand compared to the native pose of linezolid and the common binding residues of 3l, 29f, 67 and linezolid. RMSD was only used to monitor crystallographic ligands and not the ligands from the dataset. As each derivative had various modifications to the Linezolid structure, we did not think using RMSD as a metric was accurate. We have since readjusted the phrasing in the main text (now lines 453 to 457) to mention why these specific derivatives were chosen.

  1. Line 481: There is a repeated sentence “Many molecular docking studies do not consider the associating effects of water molecules and metal ions”.

Response 6: Thank you for pointing out this error. We have removed the duplicated sentence from the text.

Round 2

Reviewer 1 Report

The revised version is too lengthy and hence, must be modified to include the required contents only. As mentioned previously that the programs included in current study are not specifically used for nucleic acid docking; rather those programs are frequently used for protein-ligand or protein-protein dockings and have been validated several times before. Therefore, for assessment of docking packages, following programs which deal with nucleic acid-protein docking, must be used for evaluating ribosomal-ligand interactions:

1.      NPDock (Nucleic acid-Protein Dock) is a web server for modeling of RNA-protein and DNA-protein complex structures.

2.      HNADOCK: a nucleic acid docking server for modeling RNA/DNA–RNA/DNA 3D complex structures.

3.      HDOCK: a web server for protein–protein and protein–DNA/RNA docking based on a hybrid strategy.

4.      NLDock: a Fast Nucleic Acid–Ligand Docking Algorithm for Modeling RNA/DNA–Ligand Complexes.

Author Response

Thank you for your further comments. We have since removed pharmacophore methods (sections 2.3.7 Field template pharmacophore, and 3.8 Pharmacophore and Field Template Generation) to supplementary material (Supplementary Discussion 1, Supplementary Figure 7, and Supplementary Method 1) to help reduce the length of the manuscript and only include relevant analysis.

While web server tools like NPDock, HNADOCK, HDOCK are important RNA tools, all have been developed for either protein-protein, protein-RNA/DNA, or RNA/DNA-RNA/DNA targets and not ligand-RNA/DNA scoring. Hence, they are not appropriate and not chosen for the selected study. However, future work will focus on validating these docking tools for their applicability against rRNA-small ligand targets. We have added mention of these programs in lines 618 to 623.

For NLDOCK, we initially inquired with the authors about the availability of this program due to its ideal alignment with the structures in our study. At the time of the conduction of the docking program validation, the authors NLDock were still working on refining their code, and it was not available for public use. We have contacted them again, inquiring about the availability of this program, and we are waiting to hear from them.

Reviewer 2 Report

The authors responded to the comments in a satisfactory manner. The paper has improved on the previous version.
I recommend the publication of this paper.

Author Response

The authors thank you for taking the time to review our manuscript. Your suggestions and insights have served to strengthen and improve the clarity and quality of our manuscript.

Round 3

Reviewer 1 Report

The authors have addressed all queries, and the revised version has been improved. Therefore, the manuscript may be considered for publication in "antibiotics". However, authors may consider rephrasing the title to include "ribosome or ribosomal-oxazolidinones interactions" because the current title is too general and not specific. This can be done during the proofreading stage, and no further review is required.